# Unsupervised Data Augmentation
# for Consistency Training

**Qizhe Xie[1,2], Zihang Dai[1,2], Eduard Hovy[2], Minh-Thang Luong[1], Quoc V. Le[1]**
[1] Google Research, Brain Team, [2] Carnegie Mellon University
{qizhex, dzihang, hovy}@cs.cmu.edu, {thangluong, qvl}@google.com

## Abstract

Semi-supervised learning lately has shown much promise in improving deep learn-ing models when labeled data is scarce. Common among recent approaches is the use of consistency training on a large amount of unlabeled data to constrain model predictions to be invariant to input noise. In this work, we present a new perspective on how to effectively noise unlabeled examples and argue that the quality of noising, specifically those produced by advanced data augmentation methods, plays a crucial role in semi-supervised learning. By substituting simple noising operations with advanced data augmentation methods such as RandAug-ment and back-translation, our method brings substantial improvements across six language and three vision tasks under the same consistency training framework. On the IMDb text classification dataset, with only 20 labeled examples, our method achieves an error rate of 4.20, outperforming the state-of-the-art model trained on 25,000 labeled examples. On a standard semi-supervised learning benchmark, CIFAR-10, our method outperforms all previous approaches and achieves an error rate of 5.43 with only 250 examples. Our method also combines well with transfer learning, e.g., when finetuning from BERT, and yields improvements in high-data regime, such as ImageNet, whether when there is only 10% labeled data or when a full labeled set with 1.3M extra unlabeled examples is used.[1]

## 1  Introduction

A fundamental weakness of deep learning is that it typically requires a lot of labeled data to work well. Semi-supervised learning (SSL) [1] is one of the most promising paradigms of leveraging unlabeled data to address this weakness. The recent works in SSL are diverse but those that are based on consistency training [2, 3, 4, 5] have shown to work well on many benchmarks.

In a nutshell, consistency training methods simply regularize model predictions to be invariant to small noise applied to either input examples [6, 7, 8] or hidden states [2, 4]. This framework makes sense intuitively because a good model should be robust to any small change in an input example or hidden states. Under this framework, different methods in this category differ mostly in how and where the noise injection is applied. Typical noise injection methods are additive Gaussian noise, dropout noise or adversarial noise.

In this work, we investigate the role of noise injection in consistency training and observe that advanced data augmentation methods, specifically those work best in supervised learning [9, 10, 11, 12], also perform well in semi-supervised learning. There is indeed a strong correlation between the performance of data augmentation operations in supervised learning and their performance in consistency training. We, hence, propose to substitute the traditional noise injection methods with high quality data augmentation methods in order to improve consistency training. To emphasize the

use of better data augmentation in consistency training, we name our method Unsupervised Data Augmentation or UDA.

We evaluate UDA on a wide variety of language and vision tasks. On six text classification tasks, our method achieves significant improvements over state-of-the-art models. Notably, on IMDb, UDA with 20 labeled examples outperforms the state-of-the-art model trained on 1250x more labeled data. On standard semi-supervised learning benchmarks CIFAR-10 and SVHN, UDA outperforms all existing semi-supervised learning methods by significant margins and achieves an error rate of 5.43 and 2.72 with 250 labeled examples respectively. Finally, we also find UDA to be beneficial when there is a large amount of supervised data. For instance, on ImageNet, UDA leads to improvements of top-1 accuracy from $58.84$ to $68.78$ with $10\%$ of the labeled set and from $78.43$ to $79.05$ when we use the full labeled set and an external dataset with 1.3M unlabeled examples.

Our key contributions and findings can be summarized as follows:

- First, we show that state-of-the-art data augmentations found in supervised learning can also serve as a superior source of noise under the consistency enforcing semi-supervised framework. *See results in Table 1 and Table 2.*
- Second, we show that UDA can match and even outperform purely supervised learning that uses orders of magnitude more labeled data. *See results in Table 4 and Figure 4.*
  *State-of-the-art results for both vision and language tasks are reported in Table 3 and 4. The effectiveness of UDA across different training data sizes are highlighted in Figure 4 and 7.*
- Third, we show that UDA combines well with transfer learning, e.g., when fine-tuning from BERT (*see Table 4*), and is effective at high-data regime, e.g. on ImageNet (*see Table 5*).
- Lastly, we also provide a theoretical analysis of how UDA improves the classification performance and the corresponding role of the state-of-the-art augmentation in Section 3.

## 2   Unsupervised Data Augmentation (UDA)

In this section, we first formulate our task and then present the key method and insights behind UDA. Throughout this paper, we focus on classification problems and will use $x$ to denote the input and $y^*$ to denote its ground-truth prediction target. We are interested in learning a model $p_\theta(y \mid x)$ to predict $y^*$ based on the input $x$, where $\theta$ denotes the model parameters. Finally, we will use $p_L(x)$ and $p_U(x)$ to denote the distributions of labeled and unlabeled examples respectively and use $f^*$ to denote the perfect classifier that we hope to learn.

### 2.1   Background: Supervised Data Augmentation

Data augmentation aims at creating novel and realistic-looking training data by applying a transformation to an example, without changing its label. Formally, let $q(\hat{x} \mid x)$ be the augmentation transformation from which one can draw augmented examples $\hat{x}$ based on an original example $x$. For an augmentation transformation to be valid, it is required that any example $\hat{x} \sim q(\hat{x} \mid x)$ drawn from the distribution shares the same ground-truth label as $x$. Given a valid augmentation transformation, we can simply minimize the negative log-likelihood on augmented examples.

Supervised data augmentation can be equivalently seen as constructing an augmented labeled set from the original supervised set and then training the model on the augmented set. Therefore, the augmented set needs to provide additional inductive biases to be more effective. How to design the augmentation transformation has, thus, become critical.

In recent years, there have been significant advancements on the design of data augmentations for NLP [12], vision [10, 11] and speech [13, 14] in supervised settings. Despite the promising results, data augmentation is mostly regarded as the "cherry on the cake" which provides a steady but limited performance boost because these augmentations has so far only been applied to a set of labeled examples which is usually of a small size. Motivated by this limitation, via the consistency training framework, we extend the advancement in supervised data augmentation to semi-supervised learning where abundant unlabeled data is available.

### 2.2   Unsupervised Data Augmentation

As discussed in the introduction, a recent line of work in semi-supervised learning has been utilizing unlabeled examples to enforce smoothness of the model. The general form of these works can be summarized as follows:

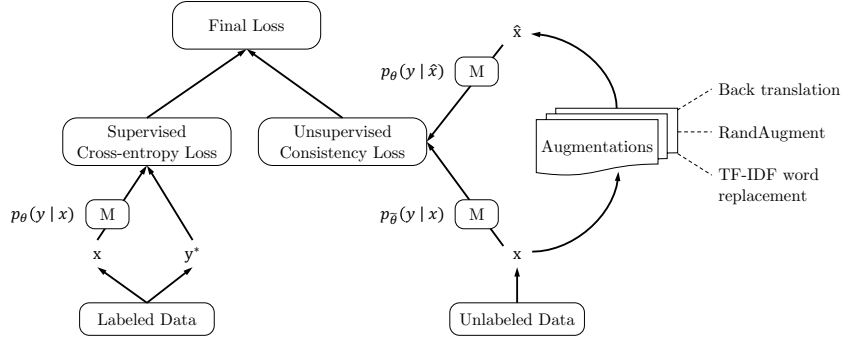

Figure 1: Training objective for UDA, where M is a model that predicts a distribution of $y$ given $x$.

- Given an input $x$, compute the output distribution $p_\theta(y \mid x)$ given $x$ and a noised version $p_\theta(y \mid x, \epsilon)$ by injecting a small noise $\epsilon$. The noise can be applied to $x$ or hidden states.
- Minimize a divergence metric between the two distributions $\mathcal{D}\left(p_\theta(y \mid x) \parallel p_\theta(y \mid x, \epsilon)\right)$.

This procedure enforces the model to be insensitive to the noise $\epsilon$ and hence smoother with respect to changes in the input (or hidden) space. From another perspective, minimizing the consistency loss gradually propagates label information from labeled examples to unlabeled ones.

In this work, we are interested in a particular setting where the noise is injected to the input $x$, i.e., $\hat{x} = q(x, \epsilon)$, as considered by prior works [7, 4, 6]. But different from existing work, we focus on the unattended question of how the form or "quality" of the noising operation $q$ can influence the performance of this consistency training framework. Specifically, to enforce consistency, prior methods generally employ simple noise injection methods such as adding Gaussian noise, simple input augmentations to noise unlabeled examples. In contrast, we hypothesize that stronger data augmentations in supervised learning can also lead to superior performance when used to noise unlabeled examples in the semi-supervised consistency training framework, since it has been shown that more advanced data augmentations that are more diverse and natural can lead to significant performance gain in the supervised setting.

Following this idea, we propose to use a rich set of state-of-the-art data augmentations verified in various supervised settings to inject noise and optimize the same consistency training objective on unlabeled examples. When jointly trained with labeled examples, we utilize a weighting factor $\lambda$ to balance the supervised cross entropy and the unsupervised consistency training loss, which is illustrated in Figure 1. Formally, the full objective can be written as follows:

$$\min_\theta \ \mathcal{J}(\theta) = \mathbb{E}_{x_1 \sim p_L(x)}\left[-\log p_\theta(f^*(x_1) \mid x_1)\right] + \lambda \mathbb{E}_{x_2 \sim p_U(x)} \mathbb{E}_{\hat{x} \sim q(\hat{x}|x_2)}\left[\text{CE}\left(p_{\tilde{\theta}}(y \mid x_2) \| p_\theta(y \mid \hat{x})\right)\right]$$
(1)

where CE denotes cross entropy, $q(\hat{x} \mid x)$ is a data augmentation transformation and $\tilde{\theta}$ is a *fixed* copy of the current parameters $\theta$ indicating that the gradient is not propagated through $\tilde{\theta}$, as suggested by VAT [6]. We set $\lambda$ to 1 for most of our experiments. In practice, in each iteration, we compute the supervised loss on a mini-batch of labeled examples and compute the consistency loss on a mini-batch of unlabeled data. The two losses are then summed for the final loss. We use a larger batch size for the consistency loss.

In the vision domain, simple augmentations including cropping and flipping are applied to labeled examples. To minimize the discrepancy between supervised training and prediction on unlabeled examples, we apply the same simple augmentations to unlabeled examples for computing $p_{\tilde{\theta}}(y \mid x)$.

**Discussion.** Before detailing the augmentation operations used in this work, we first provide some intuitions on how more advanced data augmentations can provide extra advantages over simple ones used in earlier works from three aspects:

- **Valid noise**: Advanced data augmentation methods that achieve great performance in supervised learning usually generate realistic augmented examples that share the same ground-truth labels with the original example. Thus, it is safe to encourage the consistency between predictions on the original unlabeled example and the augmented unlabeled examples.
- **Diverse noise**: Advanced data augmentation can generate a diverse set of examples since it can make large modifications to the input example without changing its label, while simple Gaussian noise only make local changes. Encouraging consistency on a diverse set of augmented examples can significantly improve the sample efficiency.

- **Targeted inductive biases**: Different tasks require different inductive biases. Data augmentation operations that work well in supervised training essentially provides the missing inductive biases.

## 2.3 Augmentation Strategies for Different Tasks

We now detail the augmentation methods, tailored for different tasks, that we use in this work.

**RandAugment for Image Classification.** We use a data augmentation method called RandAugment [15], which is inspired by AutoAugment [11]. AutoAugment uses a search method to combine all image processing transformations in the Python Image Library (PIL) to find a good augmentation strategy. In RandAugment, we do not use search, but instead uniformly sample from the same set of augmentation transformations in PIL. In other words, RandAugment is simpler and requires no labeled data as there is no need to search for optimal policies.

**Back-translation for Text Classification.** When used as an augmentation method, back-translation [16, 17] refers to the procedure of translating an existing example $x$ in language $A$ into another language $B$ and then translating it back into $A$ to obtain an augmented example $\hat{x}$. As observed by [12], back-translation can generate diverse paraphrases while preserving the semantics of the original sentences, leading to significant performance improvements in question answering. In our case, we use back-translation to paraphrase the training data of our text classification tasks.[2]

We find that the diversity of the paraphrases is important. Hence, we employ random sampling with a tunable temperature instead of beam search for the generation. As shown in Figure 2, the paraphrases generated by back-translation sentence are diverse and have similar semantic meanings. More specifically, we use WMT'14 English-French translation models (in both directions) to perform back-translation on each sentence. To facilitate future research, we have open-sourced our back-translation system together with the translation checkpoints.

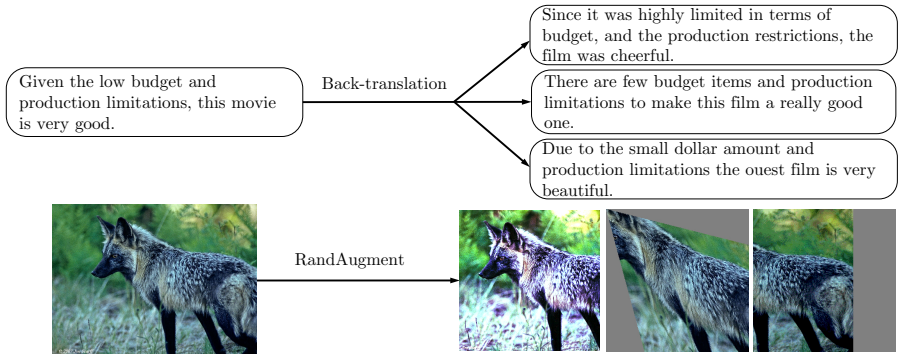

Figure 2: Augmented examples using back-translation and RandAugment.

**Word replacing with TF-IDF for Text Classification.** While back-translation is good at maintaining the global semantics of a sentence, there is little control over which words will be retained. This requirement is important for topic classification tasks, such as DBPedia, in which some keywords are more informative than other words in determining the topic. We, therefore, propose an augmentation method that replaces uninformative words with low TF-IDF scores while keeping those with high TF-IDF values. We refer readers to Appendix A.2 for a detailed description.

## 2.4 Additional Training Techniques

In this section, we present additional techniques targeting at some commonly encountered problems.

**Confidence-based masking.** We find it to be helpful to mask out examples that the current model is not confident about. Specifically, in each minibatch, the consistency loss term is computed only on examples whose highest probability among classification categories is greater than a threshold $\beta$. We set the threshold $\beta$ to a high value. Specifically, $\beta$ is set to 0.8 for CIFAR-10 and SVHN and 0.5 for ImageNet.

**Sharpening Predictions.** Since regularizing the predictions to have low entropy has been shown to be beneficial [18, 6], we sharpen predictions when computing the target distribution on unlabeled examples by using a low Softmax temperature $\tau$. When combined with confidence-based masking, the loss on unlabeled examples $\mathbb{E}_{x \sim p_U(x)} \mathbb{E}_{\hat{x} \sim q(\hat{x}|x)} \left[ \text{CE} \left( p_{\tilde{\theta}}(y \mid x) \| p_{\theta}(y \mid \hat{x}) \right) \right]$ on a minibatch $B$ is computed as:

$$\frac{1}{|B|} \sum_{x \in B} I(\max_{y'} p_{\tilde{\theta}}(y' \mid x) > \beta) \text{CE} \left( p_{\tilde{\theta}}^{(sharp)}(y \mid x) \| p_{\theta}(y \mid \hat{x}) \right)$$

$$p_{\tilde{\theta}}^{(sharp)}(y \mid x) = \frac{\exp(z_y / \tau)}{\sum_{y'} \exp(z_{y'} / \tau)}$$

where $I(\cdot)$ is the indicator function, $z_y$ is the logit of label $y$ for example $x$. We set $\tau$ to 0.4 for CIFAR-10, SVHN and ImageNet.

**Domain-relevance Data Filtering.** Ideally, we would like to make use of out-of-domain unlabeled data since it is usually much easier to collect, but the class distributions of out-of-domain data are mismatched with those of in-domain data, which can result in performance loss if directly used [19]. To obtain data relevant to the domain for the task at hand, we adopt a common technique for detecting out-of-domain data. We use our baseline model trained on the in-domain data to infer the labels of data in a large out-of-domain dataset and pick out examples that the model is most confident about. Specifically, for each category, we sort all examples based on the classified probabilities of being in that category and select the examples with the highest probabilities.

## 3   Theoretical Analysis

In this section, we theoretically analyze why UDA can improve the performance of a model and the required number of labeled examples to achieve a certain error rate. Following previous sections, we will use $f^*$ to denote the perfect classifier that we hope to learn, use $p_U$ to denote the marginal distribution of the unlabeled data and use $q(\hat{x} \mid x)$ to denote the augmentation distribution.

To make the analysis tractable, we make the following simplistic assumptions about the data augmentation transformation:

- **In-domain** augmentation: data examples generated by data augmentation have non-zero probability under $p_U$, i.e., $p_U(\hat{x}) > 0$ for $\hat{x} \sim q(\hat{x} \mid x), x \sim p_U(x)$.
- **Label-preserving** augmentation: data augmentation preserves the label of the original example, i.e., $f^*(x) = f^*(\hat{x})$ for $\hat{x} \sim q(\hat{x} \mid x), x \sim p_U(x)$.
- **Reversible** augmentation: the data augmentation operation can be reversed, i.e., if $q(\hat{x} \mid x) > 0$ then $q(x \mid \hat{x}) > 0$ .

As the first step, we hope to provide an intuitive sketch of our formal analysis. Let us define a graph $G_{p_U}$ where each node corresponds to a data sample $x \in X$ and an edge $(\hat{x}, x)$ exists in the graph *if and only if* $q(\hat{x} \mid x) > 0$. Due to the label-preserving assumption, it is easy to see that examples with different labels must reside on different components (disconnected sub-graphs) of the graph $G_{p_U}$. Hence, for an $N$-category classification problems, the graph has $N$ components (sub-graphs) when all examples within each category can be traversed by the augmentation operation. Otherwise, the graph will have more than $N$ components.

Given this construction, notice that for each component $C_i$ of the graph, as long as there is a single labeled example in the component, i.e. $(x^*, y^*) \in C_i$, one can propagate the label of the node to the rest of the nodes in $C_i$ by traversing $C_i$ via the augmentation operation $q(\hat{x} \mid x)$. More importantly, if one only performs *supervised data augmentation*, one can only propagate the label information to the directly connected neighbors of the labeled node. In contrast, performing *unsupervised data augmentation* ensures the traversal of the entire sub-graph $C_i$. This provides the first high-level intuition how UDA could help.

Taking one step further, in order to find a perfect classifier via such label propagation, it requires that there exists at least one labeled example in each component. In other words, the number of components lower bounds the minimum amount of labeled examples needed to learn a perfect classifier. Importantly, number of components is actually decided by the quality of the augmentation operation: an ideal augmentation should be able to reach all other examples of the same category given a starting instance. This well matches our discussion of the benefits of state-of-the-art data

augmentation methods in generating more diverse examples. Effectively, the augmentation diversity leads to more neighbors for each node, and hence reduces the number of components in a graph.

Since supervised data augmentation only propagates the label information to the directly connected neighbors of the labeled nodes. Advanced data augmentation that has a high accuracy must lead to a graph where each node has more neighbors. Effectively, such a graph has more edges and better connectivity. Hence, it is also more likely that this graph will have a smaller number of components. To further illustrate this intuition, in Figure 3, we provide a comparison between different algorithms.

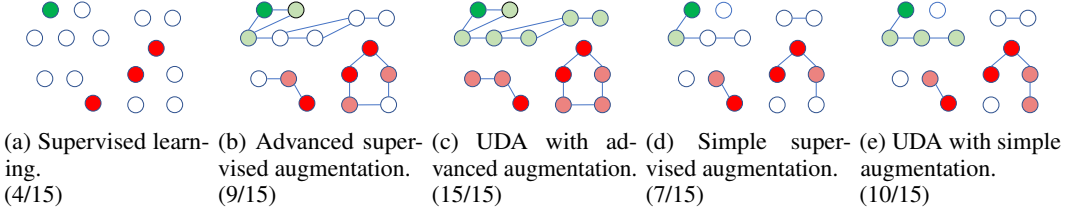

(a) Supervised learning. (4/15)  (b) Advanced supervised augmentation. (9/15)  (c) UDA with advanced augmentation. (15/15)  (d) Simple supervised augmentation. (7/15)  (e) UDA with simple augmentation. (10/15)

Figure 3: Prediction results of different settings, where green and red nodes are labeled nodes, white nodes are unlabeled nodes whose labels cannot be determined and light green nodes and light red nodes are unlabeled nodes whose labels can be correctly determined. The accuracy of different settings are shown in $(\cdot)$.

With the intuition described, we state our formal results. Without loss of generality, assume there are $k$ components in the graph. For each component $C_i (i = 1, \ldots, k)$, let $P_i$ be the total probability mass that an observed labeled example fall into the $i$-th component, i.e., $P_i = \sum_{x \in C_i} p_L(x)$. The following theorem characterizes the relationship between UDA error rate and the amount of labeled examples.

**Theorem 1.** *Under UDA, let $Pr(\mathcal{A})$ denote the probability that the algorithm cannot infer the label of a new test example given $m$ labeled examples from $P_L$. $Pr(\mathcal{A})$ is given by*

$$Pr(\mathcal{A}) = \sum_i P_i (1 - P_i)^m.$$

*In addition, $O(k/\epsilon)$ labeled examples can guarantee an error rate of $O(\epsilon)$, i.e.,*

$$m = O(k/\epsilon) \implies Pr(\mathcal{A}) = O(\epsilon).$$

*Proof.* Please see Appendix. C for details. □

From the theorem, we can see the number of components, i.e. $k$, directly governs the amount of labeled data required to reach a desired performance. As we have discussed above, the number of components effectively relies on the quality of an augmentation function, where better augmentation functions result in fewer components. This echoes our discussion of the benefits of state-of-the-art data augmentation operations in generating more diverse examples. Hence, with state-of-the-art augmentation operations, UDA is able to achieve good performance using fewer labeled examples.

## 4 Experiments

In this section, we evaluate UDA on a variety of language and vision tasks. For language, we rely on six text classification benchmark datasets, including IMDb, Yelp-2, Yelp-5, Amazon-2 and Amazon-5 sentiment classification and DBPedia topic classification [20, 21]. For vision, we employ two smaller datasets CIFAR-10 [22], SVHN [23], which are often used to compare semi-supervised algorithms, as well as ImageNet [24] of a larger scale to test the scalability of UDA. For ablation studies and experiment details, we refer readers to Appendix B and Appendix E.

### 4.1 Correlation between Supervised and Semi-supervised Performances

As the first step, we try to verify the fundamental idea of UDA, i.e., there is a positive correlation of data augmentation's effectiveness in supervised learning and semi-supervised learning. Based on Yelp-5 (a language task) and CIFAR-10 (a vision task), we compare the performance of different data augmentation methods in either fully supervised or semi-supervised settings. For Yelp-5, apart from back-translation, we include a simpler method Switchout [25] which replaces a token with a random

| Augmentation (# Sup examples) | Sup (50k) | Semi-Sup (4k) |
|---|---|---|
| Crop & flip | 5.36 | 10.94 |
| Cutout | 4.42 | 5.43 |
| RandAugment | **4.23** | **4.32** |

Table 1: Error rates on CIFAR-10.

| Augmentation (# Sup examples) | Sup (650k) | Semi-sup (2.5k) |
|---|---|---|
| ✗ | 38.36 | 50.80 |
| Switchout | 37.24 | 43.38 |
| Back-translation | **36.71** | **41.35** |

Table 2: Error rate on Yelp-5.

token uniformly sampled from the vocabulary. For CIFAR-10, we compare RandAugment with two simpler methods: (1) cropping & flipping augmentation and (2) Cutout.

Based on this setting, Table 1 and Table 2 exhibit a strong correlation of an augmentation's effectiveness between supervised and semi-supervised settings. This validates our idea of stronger data augmentations found in supervised learning can always lead to more gains when applied to the semi-supervised learning settings.

## 4.2 Algorithm Comparison on Vision Semi-supervised Learning Benchmarks

With the correlation established above, the next question we ask is how well UDA performs compared to existing semi-supervised learning algorithms. To answer the question, we focus on the most commonly used semi-supervised learning benchmarks CIFAR-10 and SVHN.

**Vary the size of labeled data.** Firstly, we follow the settings in [19] and employ Wide-ResNet-28-2 [26, 27] as the backbone model and evaluate UDA with varied supervised data sizes. Specifically, we compare UDA with two highly competitive baselines: (1) Virtual adversarial training (VAT) [6], an algorithm that generates adversarial Gaussian noise on input, and (2) MixMatch [28], a parallel work that combines previous advancements in semi-supervised learning. The comparison is shown in Figure 4 with two key observations.

- First, UDA consistently outperforms the two baselines given different sizes of labeled data.
- Moreover, the performance difference between UDA and VAT shows the superiority of data augmentation based noise. The difference of UDA and VAT is essentially the noise process. While the noise produced by VAT often contain high-frequency artifacts that do not exist in real images, data augmentation mostly generates diverse and realistic images.

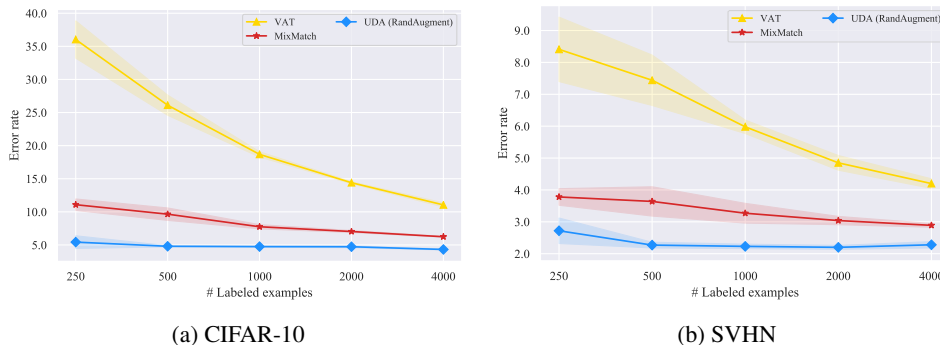

(a) CIFAR-10        (b) SVHN

Figure 4: Comparison with two semi-supervised learning methods on CIFAR-10 and SVHN with varied number of labeled examples.

**Vary model architecture.** Next, we directly compare UDA with previously published results under different model architectures. Following previous work, 4k and 1k labeled examples are used for CIFAR-10 and SVHN respectively. As shown in Table 3, given the same architecture, UDA outperforms all published results by significant margins and nearly matches the fully supervised performance, which uses 10x more labeled examples. This shows the huge potential of state-of-the-art data augmentations under the consistency training framework in the vision domain.

## 4.3 Evaluation on Text Classification Datasets

Next, we further evaluate UDA in the language domain. Moreover, in order to test whether UDA can be combined with the success of unsupervised representation learning, such as BERT [34], we further consider four initialization schemes: (a) random Transformer; (b) BERT$_{\text{BASE}}$; (c) BERT$_{\text{LARGE}}$; (d)

| Method | Model | # Param | CIFAR-10 (4k) | SVHN (1k) |
|---|---|---|---|---|
| $\Pi$-Model [4] | Conv-Large | 3.1M | $12.36 \pm 0.31$ | $4.82 \pm 0.17$ |
| Mean Teacher [5] | Conv-Large | 3.1M | $12.31 \pm 0.28$ | $3.95 \pm 0.19$ |
| VAT + EntMin [6] | Conv-Large | 3.1M | $10.55 \pm 0.05$ | $3.86 \pm 0.11$ |
| SNTG [29] | Conv-Large | 3.1M | $10.93 \pm 0.14$ | $3.86 \pm 0.27$ |
| ICT [30] | Conv-Large | 3.1M | $7.29 \pm 0.02$ | $3.89 \pm 0.04$ |
| Pseudo-Label [31] | WRN-28-2 | 1.5M | $16.21 \pm 0.11$ | $7.62 \pm 0.29$ |
| LGA + VAT [32] | WRN-28-2 | 1.5M | $12.06 \pm 0.19$ | $6.58 \pm 0.36$ |
| ICT [30] | WRN-28-2 | 1.5M | $7.66 \pm 0.17$ | $3.53 \pm 0.07$ |
| MixMatch [28] | WRN-28-2 | 1.5M | $6.24 \pm 0.06$ | $2.89 \pm 0.06$ |
| Mean Teacher [5] | Shake-Shake | 26M | $6.28 \pm 0.15$ | - |
| Fast-SWA [33] | Shake-Shake | 26M | 5.0 | - |
| MixMatch [28] | WRN | 26M | $4.95 \pm 0.08$ | - |
| UDA (RandAugment) | WRN-28-2 | 1.5M | $4.32 \pm 0.08$ | $\mathbf{2.23 \pm 0.07}$ |
| UDA (RandAugment) | Shake-Shake | 26M | 3.7 | - |
| UDA (RandAugment) | PyramidNet | 26M | **2.7** | - |

Table 3: Comparison between methods using different models where PyramidNet is used with ShakeDrop regularization. On CIFAR-10, with only 4,000 labeled examples, UDA matches the performance of fully supervised Wide-ResNet-28-2 and PyramidNet+ShakeDrop, where they have an error rate of 5.4 and 2.7 respectively when trained on 50,000 examples without RandAugment. On SVHN, UDA also matches the performance of our fully supervised model trained on 73,257 examples without RandAugment, which has an error rate of 2.84.

BERT$_{\text{FINETUNE}}$: BERT$_{\text{LARGE}}$ fine-tuned on in-domain unlabeled data[3]. Under each of these four initialization schemes, we compare the performances with and without UDA.

| **Fully supervised baseline** | | | | | | |
|---|---|---|---|---|---|---|
| **Datasets** (# Sup examples) | IMDb (25k) | Yelp-2 (560k) | Yelp-5 (650k) | Amazon-2 (3.6m) | Amazon-5 (3m) | DBpedia (560k) |
| Pre-BERT SOTA | *4.32* | 2.16 | 29.98 | 3.32 | 34.81 | 0.70 |
| BERT$_{\text{LARGE}}$ | 4.51 | *1.89* | *29.32* | *2.63* | *34.17* | *0.64* |

| **Semi-supervised setting** | | | | | | | |
|---|---|---|---|---|---|---|---|
| **Initialization** | **UDA** | IMDb (20) | Yelp-2 (20) | Yelp-5 (2.5k) | Amazon-2 (20) | Amazon-5 (2.5k) | DBpedia (140) |
| Random | ✗ | 43.27 | 40.25 | 50.80 | 45.39 | 55.70 | 41.14 |
|  | ✓ | 25.23 | 8.33 | 41.35 | 16.16 | 44.19 | 7.24 |
| BERT$_{\text{BASE}}$ | ✗ | 18.40 | 13.60 | 41.00 | 26.75 | 44.09 | 2.58 |
|  | ✓ | 5.45 | 2.61 | 33.80 | 3.96 | 38.40 | 1.33 |
| BERT$_{\text{LARGE}}$ | ✗ | 11.72 | 10.55 | 38.90 | 15.54 | 42.30 | 1.68 |
|  | ✓ | 4.78 | 2.50 | 33.54 | 3.93 | 37.80 | 1.09 |
| BERT$_{\text{FINETUNE}}$ | ✗ | 6.50 | 2.94 | 32.39 | 12.17 | 37.32 | - |
|  | ✓ | **4.20** | **2.05** | **32.08** | **3.50** | **37.12** | - |

Table 4: Error rates on text classification datasets. In the fully supervised settings, the pre-BERT SOTAs include ULMFiT [35] for Yelp-2 and Yelp-5, DPCNN [36] for Amazon-2 and Amazon-5, Mixed VAT [37] for IMDb and DBPedia. All of our experiments use a sequence length of 512.

The results are presented in Table 4 where we would like to emphasize three observations:

- First, even with very few labeled examples, UDA can offer decent or even competitive performances compared to the SOTA model trained with full supervised data. Particularly, on binary sentiment analysis tasks, with only 20 supervised examples, UDA outperforms the previous SOTA trained with full supervised data on IMDb and is competitive on Yelp-2 and Amazon-2.

- Second, UDA is complementary to transfer learning / representation learning. As we can see, when initialized with BERT and further finetuned on in-domain data, UDA can still significantly reduce the error rate from 6.50 to 4.20 on IMDb.

- Finally, we also note that for five-category sentiment classification tasks, there still exists a clear gap between UDA with 500 labeled examples per class and BERT trained on the entire supervised

set. Intuitively, five-category sentiment classifications are much more difficult than their binary counterparts. This suggests a room for further improvement in the future.

### 4.4 Scalability Test on the ImageNet Dataset

Then, to evaluate whether UDA can scale to problems with a large scale and a higher difficulty, we now turn to the ImageNet dataset with ResNet-50 being the underlying architecture. Specifically, we consider two experiment settings with different natures:

- We use 10% of the supervised data of ImageNet while using all other data as unlabeled data. As a result, the unlabeled exmaples are entirely in-domain.
- In the second setting, we keep all images in ImageNet as supervised data. Then, we use the domain-relevance data filtering method to filter out 1.3M images from JFT [38, 39]. Hence, the unlabeled set is not necessarily in-domain.

The results are summarized in Table 5. In both 10% and the full data settings, UDA consistently brings significant gains compared to the supervised baseline. This shows UDA is not only able to scale but also able to utilize out-of-domain unlabeled examples to improve model performance. In parallel to our work, S4L [40] and CPC [41] also show significant improvements on ImageNet.

| Methods | SSL | 10% | 100% |
|---|---|---|---|
| ResNet-50 w. RandAugment | ✗ | 55.09 / 77.26 58.84 / 80.56 | 77.28 / 93.73 78.43 / 94.37 |
| UDA (RandAugment) | ✓ | **68.78 / 88.80** | **79.05 / 94.49** |

Table 5: Top-1 / top-5 accuracy on ImageNet with 10% and 100% of the labeled set. We use image size 224 and 331 for the 10% and 100% experiments respectively.

## 5 Related Work

Existing works in consistency training does make use of data augmentation [4, 7]; however, they only apply weak augmentation methods such as random translations and cropping. In parallel to our work, ICT [30] and MixMatch [28] also show improvements for semi-supervised learning. These methods employ mixup [42] on top of simple augmentations such as flipping and cropping; instead, UDA emphasizes on the use of state-of-the-art data augmentations, leading to significantly better results on CIFAR-10 and SVHN. In addition, UDA is also applicable to language domain and can also scale well to more challenging vision datasets, such as ImageNet.

Other works in the consistency training family mostly differ in how the noise is defined: Pseudo-ensemble [2] directly applies Gaussian noise and Dropout noise; VAT [6, 43] defines the noise by approximating the direction of change in the input space that the model is most sensitive to; Cross-view training [8] masks out part of the input data. Apart from enforcing consistency on the input examples and the hidden representations, another line of research enforces consistency on the model parameter space. Works in this category include Mean Teacher [5], fast-Stochastic Weight Averaging [33] and Smooth Neighbors on Teacher Graphs [29]. For a complete version of related work, please refer to Appendix D.

## 6 Conclusion

In this paper, we show that data augmentation and semi-supervised learning are well connected: better data augmentation can lead to significantly better semi-supervised learning. Our method, UDA, employs state-of-the-art data augmentation found in supervised learning to generate diverse and realistic noise and enforces the model to be consistent with respect to these noise. For text, UDA combines well with representation learning, e.g., BERT. For vision, UDA outperforms prior works by a clear margin and nearly matches the performance of the fully supervised models trained on the full labeled sets which are one order of magnitude larger. We hope that UDA will encourage future research to transfer advanced supervised augmentation to semi-supervised setting for different tasks.

## Acknowledgements

We want to thank Hieu Pham, Adams Wei Yu, Zhilin Yang and Ekin Dogus Cubuk for their tireless help to the authors on different stages of this project and thank Colin Raffel for pointing out the connections between our work and previous works. We also would like to thank Olga Wichrowska, Barret Zoph, Jiateng Xie, Guokun Lai, Yulun Du, Chen Dan, David Berthelot, Avital Oliver, Trieu Trinh, Ran Zhao, Ola Spyra, Brandon Yang, Daiyi Peng, Andrew Dai, Samy Bengio, Jeff Dean and the Google Brain team for insightful discussions and support to the work. Lastly, we thank anonymous reviewers for their valueable feedbacks.

## Broader Impact

This work show that it is possible to achieve great performance with limited labeled data. Hence groups/institutes with limited budgets for annotating data may benefit from this research. To the best of our knowledge, nobody will be put at disadvantage from this research. Our method does not leverage biases in the data. Our tasks include standard benchmarks such as IMDb, CIFAR-10, SVHN and ImageNet.

## Footnotes

[1]Code is available at https://github.com/google-research/uda.

[2]We also note that while translation uses a labeled dataset, the translation task itself is quite distinctive from a text classification task and does not make use of any text classification label. In addition, back-translation is a general data augmentation method that can be applied to many tasks with the same model checkpoints.

[3]One exception is that we do not pursue BERT$_{\text{FINETUNE}}$ on DBPedia as fine-tuning BERT on DBPedia does not yield further performance gain. This is probably due to the fact that DBPedia is based on Wikipedia while BERT is already trained on the whole Wikipedia corpus.

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
