[Supplementary Material]

# A  Extended Method Details

In this section, we present some additional details used in our method. We introduce Training Signal Annealing in Appendix A.1 and details for augmentation strategies in Appendix A.2.

## A.1  Training Signal Annealing for Low-data Regime

In semi-supervised learning, we often encounter a situation where there is a huge gap between the amount of unlabeled data and that of labeled data. Hence, the model often quickly overfits the limited amount of labeled data while still underfitting the unlabeled data. To tackle this difficulty, we introduce a new training technique, called Training Signal Annealing (TSA), which gradually releases the "training signals" of the labeled examples as training progresses. Intuitively, we only utilize a labeled example if the model's confidence on that example is lower than a predefined threshold which increases according to a schedule. Specifically, at training step $t$, if the model's predicted probability for the correct category $p_\theta(y^* \mid x)$ is higher than a threshold $\eta_t$, we remove that example from the loss function. Suppose $K$ is the number of categories, by gradually increasing $\eta_t$ from $\frac{1}{K}$ to 1, the threshold $\eta_t$ serves as a ceiling to prevent over-training on easy labeled examples.

We consider three increasing schedules of $\eta_t$ with different application scenarios. Let $T$ be the total number of training steps, the three schedules are shown in Figure 5. Intuitively, when the model is prone to overfit, e.g., when the problem is relatively easy or the number of labeled examples is very limited, the exp-schedule is most suitable as the supervised signal is mostly released at the end of training. In contrast, when the model is less likely to overfit (e.g., when we have abundant labeled examples or when the model employs effective regularization), the log-schedule can serve well.

Figure 5: Three schedules of TSA. We set $\eta_t = \alpha_t * (1 - \frac{1}{K}) + \frac{1}{K}$. $\alpha_t$ is set to $1 - \exp(-\frac{t}{T} * 5)$, $\frac{t}{T}$ and $\exp((\frac{t}{T} - 1) * 5)$ for the log, linear and exp schedules.

## A.2  Extended Augmentation Strategies for Different Tasks

**Discussion on Trade-off Between Diversity and Validity for Data Augmentation.** Despite that state-of-the-art data augmentation methods can generate diverse and valid augmented examples as discussed in section 2.2, there is a trade-off between diversity and validity since diversity is achieved by changing a part of the original example, naturally leading to the risk of altering the ground-truth label. We find it beneficial to tune the trade-off between diversity and validity for data augmentation methods. For text classification, we tune the temperature of random sampling. On the one hand, when we use a temperature of 0, decoding by random sampling degenerates into greedy decoding and generates perfectly valid but identical paraphrases. On the other hand, when we use a temperature of 1, random sampling generates very diverse but barely readable paraphrases. We find that setting the Softmax temperature to $0.7, 0.8$ or $0.9$ leads to the best performances.

**RandAugment Details.** In our implementation of RandAugment, each sub-policy is composed of two operations, where each operation is represented by the transformation name, probability, and magnitude that is specific to that operation. For example, a sub-policy can be [(Sharpness, 0.6, 2), (Posterize, 0.3, 9)].

For each operation, we randomly sample a transformation from 15 possible transformations, a magnitude in $[1, 10]$ and fix the probability to $0.5$. Specifically, we sample from the following 15 transformations: Invert, Cutout, Sharpness, AutoContrast, Posterize, ShearX, TranslateX, TranslateY, ShearY, Rotate, Equalize, Contrast, Color, Solarize, Brightness. We find this setting to work well in

our first try and did not tune the magnitude range and the probability. Tuning these hyperparameters might result in further gains in accuracy.

**TF-IDF based word replacing Details.** Ideally, we would like the augmentation method to generate both diverse and valid examples. Hence, the augmentation is designed to retain keywords and replace uninformative words with other uninformative words. We use BERT's word tokenizer since BERT first tokenizes sentences into a sequence of words and then tokenize words into subwords although the model uses subwords as input.

Specifically, Suppose $\text{IDF}(w)$ is the IDF score for word $w$ computed on the whole corpus, and $\text{TF}(w)$ is the TF score for word $w$ in a sentence. We compute the TF-IDF score as $\text{TFIDF}(w) = \text{TF}(w)\text{IDF}(w)$. Suppose the maximum TF-IDF score in a sentence $x$ is $C = \max_i \text{TFIDF}(x_i)$. To make the probability of having a word replaced to negatively correlate with its TF-IDF score, we set the probability to $\min(p(C - \text{TFIDF}(x_i))/Z, 1)$, where $p$ is a hyperparameter that controls the magnitude of the augmentation and $Z = \sum_i (C - \text{TFIDF}(x_i))/|x|$ is the average score. $p$ is set to 0.7 for experiments on DBPedia.

When a word is replaced, we sample another word from the whole vocabulary for the replacement. Intuitively, the sampled words should not be keywords to prevent changing the ground-truth labels of the sentence. To measure if a word is keyword, we compute a score of each word on the whole corpus. Specifically, we compute the score as $S(w) = \text{freq}(w)\text{IDF}(w)$ where $\text{freq}(w)$ is the frequency of word $w$ on the whole corpus. We set the probability of sampling word $w$ as $(\max_{w'} S(w') - S(w))/Z'$ where $Z' = \sum_w \max_{w'} S(w') - S(w)$ is a normalization term.

# B    Extended Experiments

## B.1    Ablation Studies

**Ablation Studies for Unlabeled Data Size**   Here we present an ablation study for unlabeled data sizes. As shown in Table 6 and Table 7, given the same number of labeled examples, reducing the number of unsupervised examples clearly leads to worse performance. In fact, having abundant unsupervised examples is more important than having more labeled examples since reducing the unlabeled data amount leads to worse performance than reducing the labeled data by the same ratio.

| # Unsup / # Sup | 250 | 500 | 1,000 | 2,000 | 4,000 |
|---|---|---|---|---|---|
| 50,000 | $5.43 \pm 0.96$ | $4.80 \pm 0.09$ | $4.75 \pm 0.10$ | $4.73 \pm 0.14$ | $4.32 \pm 0.08$ |
| 20,000 | $11.01 \pm 1.01$ | $9.46 \pm 0.14$ | $8.57 \pm 0.14$ | $7.65 \pm 0.17$ | $7.31 \pm 0.24$ |
| 10,000 | $23.17 \pm 0.71$ | $18.43 \pm 0.43$ | $15.46 \pm 0.58$ | $12.52 \pm 0.13$ | $10.32 \pm 0.20$ |
| 5,000 | $35.41 \pm 0.75$ | $28.35 \pm 0.60$ | $22.06 \pm 0.71$ | $17.36 \pm 0.15$ | $13.19 \pm 0.12$ |

Table 6: Error rate (%) for CIFAR-10 with different amounts of labeled data and unlabeled data.

| # Unsup / # Sup | 250 | 500 | 1,000 | 2,000 | 4,000 |
|---|---|---|---|---|---|
| 73,257 | $2.72 \pm 0.40$ | $2.27 \pm 0.09$ | $2.23 \pm 0.07$ | $2.20 \pm 0.06$ | $2.28 \pm 0.10$ |
| 20,000 | $5.59 \pm 0.74$ | $4.43 \pm 0.15$ | $3.81 \pm 0.11$ | $3.86 \pm 0.14$ | $3.64 \pm 0.20$ |
| 10,000 | $17.13 \pm 12.85$ | $7.59 \pm 1.01$ | $5.76 \pm 0.29$ | $5.17 \pm 0.12$ | $5.40 \pm 0.12$ |
| 5,000 | $31.58 \pm 7.39$ | $12.66 \pm 0.81$ | $6.28 \pm 0.25$ | $8.35 \pm 0.36$ | $7.76 \pm 0.28$ |

Table 7: Error rate (%) for SVHN with different amounts of labeled data and unlabeled data.

**Ablations Studies on RandAugment**   We hypothesize that the success of RandAugment should be credited to the diversity of the augmentation transformations, since RandAugment works very well for multiple different datasets while it does not require a search algorithm to find out the most effective policies. To verify this hypothesis, we test UDA's performance when we restrict the number of possible transformations used in RandAugment. As shown in Figure 6, the performance gradually improves as we use more augmentation transformations.

Figure 6: Error rate of UDA on CIFAR-10 with different numbers of possible transformations in RandAugment. UDA achieves lower error rate when we increase the number of possible transformations, which demonstrates the importance of a rich set of augmentation transformations.

**Ablation Studies for TSA** We study the effect of TSA on Yelp-5 where we have 2.5k labeled examples and 6m unlabeled examples. We use a randomly initialized transformer in this study to rule out factors of having a pre-trained representation.

As shown in Table 8, on Yelp-5, where there is a lot more unlabeled data than labeled data, TSA reduces the error rate from 50.81 to 41.35 when compared to the baseline without TSA. More specifically, the best performance is achieved when we choose to postpone releasing the supervised training signal to the end of the training, i.e, exp-schedule leads to the best performance.

| TSA schedule | Yelp-5 |
|---|---|
| ✗ | 50.81 |
| log-schedule | 49.06 |
| linear-schedule | 45.41 |
| exp-schedule | **41.35** |

Table 8: Ablation study for Training Signal Annealing (TSA) on Yelp-5 and CIFAR-10. The shown numbers are error rates.

## B.2 More Results on CIFAR-10, SVHN and Text Classification Datasets

**Results with varied label set sizes on CIFAR-10** In Table 9, we show results for compared methods of Figure 4a and results of Pseudo-Label [33], Π-Model [32], Mean Teacher [58]. Fully supervised learning using 50,000 examples achieves an error rate of 4.23 and 5.36 with or without RandAugment. The performance of the baseline models are reported by MixMatch [3].

To make sure that the performance reported by MixMatch and our results are comparable, we reimplement MixMatch in our codebase and find that the results in the original paper is comparable but slightly better than our reimplementation, which results in a more competitive comparison for UDA. For example, our reimplementation of MixMatch achieves an error rate of $7.00 \pm 0.59$ and $7.39 \pm 0.11$ with 4,000 and 2,000 examples.

| Methods / # Sup | 250 | 500 | 1,000 | 2,000 | 4,000 |
|---|---|---|---|---|---|
| Pseudo-Label | $49.98 \pm 1.17$ | $40.55 \pm 1.70$ | $30.91 \pm 1.73$ | $21.96 \pm 0.42$ | $16.21 \pm 0.11$ |
| Π-Model | $53.02 \pm 2.05$ | $41.82 \pm 1.52$ | $31.53 \pm 0.98$ | $23.07 \pm 0.66$ | $17.41 \pm 0.37$ |
| Mean Teacher | $47.32 \pm 4.71$ | $42.01 \pm 5.86$ | $17.32 \pm 4.00$ | $12.17 \pm 0.22$ | $10.36 \pm 0.25$ |
| VAT | $36.03 \pm 2.82$ | $26.11 \pm 1.52$ | $18.68 \pm 0.40$ | $14.40 \pm 0.15$ | $11.05 \pm 0.31$ |
| MixMatch | $11.08 \pm 0.87$ | $9.65 \pm 0.94$ | $7.75 \pm 0.32$ | $7.03 \pm 0.15$ | $6.24 \pm 0.06$ |
| UDA (RandAugment) | $\mathbf{5.43 \pm 0.96}$ | $\mathbf{4.80 \pm 0.09}$ | $\mathbf{4.75 \pm 0.10}$ | $\mathbf{4.73 \pm 0.14}$ | $\mathbf{4.32 \pm 0.08}$ |

Table 9: Error rate (%) for CIFAR-10.

**Results with varied label set sizes on SVHN**   In Table 10, we similarly show results for compared methods of Figure 4b and results of methods mentioned above. Fully supervised learning using 73,257 examples achieves an error rate of 2.28 and 2.84 with or without RandAugment. The performance of the baseline models are reported by MixMatch [3]. Our reimplementation of MixMatch also resulted in comparable but higher error rates than the reported ones.

| Methods / # Sup | 250 | 500 | 1,000 | 2,000 | 4,000 |
|---|---|---|---|---|---|
| Pseudo-Label | $21.16 \pm 0.88$ | $14.35 \pm 0.37$ | $10.19 \pm 0.41$ | $7.54 \pm 0.27$ | $5.71 \pm 0.07$ |
| $\Pi$-Model | $17.65 \pm 0.27$ | $11.44 \pm 0.39$ | $8.60 \pm 0.18$ | $6.94 \pm 0.27$ | $5.57 \pm 0.14$ |
| Mean Teacher | $6.45 \pm 2.43$ | $3.82 \pm 0.17$ | $3.75 \pm 0.10$ | $3.51 \pm 0.09$ | $3.39 \pm 0.11$ |
| VAT | $8.41 \pm 1.01$ | $7.44 \pm 0.79$ | $5.98 \pm 0.21$ | $4.85 \pm 0.23$ | $4.20 \pm 0.15$ |
| MixMatch | $3.78 \pm 0.26$ | $3.64 \pm 0.46$ | $3.27 \pm 0.31$ | $3.04 \pm 0.13$ | $2.89 \pm 0.06$ |
| UDA (RandAugment) | $\mathbf{2.72 \pm 0.40}$ | $\mathbf{2.27 \pm 0.09}$ | $\mathbf{2.23 \pm 0.07}$ | $\mathbf{2.20 \pm 0.06}$ | $\mathbf{2.28 \pm 0.10}$ |

Table 10: Error rate (%) for SVHN.

(a) IMDb

(b) Yelp-2

Figure 7: Accuracy on IMDb and Yelp-2 with different number of labeled examples. In the large-data regime, with the full training set of IMDb, UDA also provides robust gains.

**Experiments on Text Classification with Varied Label Set Sizes**   We also try different data sizes on text classification tasks . As show in Figure 7, UDA leads to consistent improvements across all labeled data sizes on IMDb and Yelp-2.

## C   Proof for Theoretical Analysis

Here, we provide a full proof for Theorem 1.

**Theorem 1.** *Under UDA, let $Pr(\mathcal{A})$ denote the probability that the algorithm cannot infer the label of a new test example given $m$ labeled examples from $P_L$. $Pr(\mathcal{A})$ is given by*

$$Pr(\mathcal{A}) = \sum_i P_i(1 - P_i)^m.$$

*In addition, $O(k/\epsilon)$ labeled examples can guarantee an error rate of $O(\epsilon)$, i.e.,*

$$m = O(k/\epsilon) \implies Pr(\mathcal{A}) = O(\epsilon).$$

*Proof.* Let $x'$ be the sampled test example. Then the probability of event $\mathcal{A}$ is

$$Pr(\mathcal{A}) = \sum_i Pr(\mathcal{A} \text{ and } x' \in C_i) = \sum_i P_i(1 - P_i)^m$$

To bound the probability, we would like to find the maximum value of $\sum_i P_i(1 - P_i)^m$. We can define the following optimization function:

$$\min_P -\sum_{c_i} P_i(1 - P_i)^m$$

$$\text{s.t.} \sum_{c_i} P_i = 1$$

The problem is a convex optimization problem and we can construct its the Lagrangian dual function:

$$\mathcal{L} = \sum_i P_i(1 - P_i)^m - \lambda(\sum_i P_i - 1)$$

Using the KKT condition, we can take derivatives to $P_i$ and set it to zero. Then we have

$$\lambda = (1 - mP_i)(1 - P_i)^{m-1}$$

Hence $P_i = P_j$ for any $i \neq j$. Using the fact that $\sum_i P_i = 1$, we have

$$P_i = \frac{1}{k}$$

Plugging the result back into $Pr(\mathcal{A}) = \sum_i P_i(1 - P_i)^m$, we have

$$Pr(\mathcal{A}) \leq (1 - \frac{1}{k})^m = \exp(m\log(1 - \frac{1}{k})) \leq \exp(-\frac{m}{k})$$

Hence when $m = O(\frac{k}{\epsilon})$, we have

$$Pr(\mathcal{A}) = O(\epsilon)$$

$\square$

## D   Extended Related Work

**Semi-supervised Learning.** Due to the long history of semi-supervised learning (SSL), we refer readers to [5] for a general review. More recently, many efforts have been made to renovate classic ideas into deep neural instantiations. For example, graph-based label propagation [72] has been extended to neural methods via graph embeddings [62, 63] and later graph convolutions [28]. Similarly, with the variational auto-encoding framework and reinforce algorithm, classic graphical models based SSL methods with target variable being latent can also take advantage of deep architectures [27, 36, 64]. Besides the direct extensions, it was found that training neural classifiers to classify out-of-domain examples into an additional class [53] works very well in practice. Later, Dai et al. [12] shows that this can be seen as an instantiation of low-density separation.

Apart from enforcing consistency on the noised input examples and the hidden representations, another line of research enforces consistency under different model parameters, which is complementary to our method. For example, Mean Teacher [58] maintains a teacher model with parameters being the ensemble of a student model's parameters and enforces the consistency between the predictions of the two models. Recently, fast-SWA [1] improves Mean Teacher by encouraging the model to explore a diverse set of plausible parameters. In addition to parameter-level consistency, SNTG [35] also enforces input-level consistency by constructing a similarity graph between unlabeled examples.

**Data Augmentation.** Also related to our work is the field of data augmentation research. Besides the conventional approaches and two data augmentation methods mentioned in Section 2.1, a recent approach MixUp [70] goes beyond data augmentation from a single data point and performs interpolation of data pairs to achieve augmentation. Recently, it has been shown that data augmentation can be regarded as a kind of explicit regularization methods similar to Dropout [21].

**Diverse Back Translation.** Diverse paraphrases generated by back-translation has been a key component in the significant performance improvements in our text classification experiments. We use random sampling instead of beam search for decoding similar to [15]. There are also recent

works on generating diverse translations [19, 55, 29] that might lead to further improvements when used as data augmentations.

**Unsupervised Representation Learning.** Apart from semi-supervised learning, unsupervised representation learning offers another way to utilize unsupervised data. Collobert and Weston [8] demonstrated that word embeddings learned by language modeling can improve the performance significantly on semantic role labeling. Later, the pre-training of word embeddings was simplified and substantially scaled in Word2Vec [39] and Glove [46]. More recently, pre-training using language modeling and denoising auto-encoding has been shown to lead to significant improvements on many tasks in the language domain [11, 47, 48, 23, 14]. There is also a growing interest in self-supervised learning for vision [69, 20, 59].

**Consistency Training in Other Domains.** Similar ideas of consistency training has also been applied in other domains. For example, recently, enforcing adversarial consistency on unsupervised data has also been shown to be helpful in adversarial robustness [57, 68, 42, 4]. Enforcing consistency w.r.t data augmentation has also been shown to work well for representation learning [24, 65]. Invariant representation learning [34, 52] applies the consistency loss not only to the predicted distributions but also to representations and has been shown significant improvements on speech recognition.

# E    Experiment Details

## E.1    Text Classifications

**Datasets.** In our semi-supervised setting, we randomly sampled labeled examples from the full supervised set[4] and use the same number of examples for each category. For unlabeled data, we use the whole training set for DBPedia, the concatenation of the training set and the unlabeled set for IMDb and external data for Yelp-2, Yelp-5, Amazon-2 and Amazon-5 [38][5]. Note that for Yelp and Amazon based datasets, the label distribution of the unlabeled set might not match with that of labeled datasets since there are different number of examples in different categories. Nevertheless, we find it works well to use all the unlabeled data.

**Preprocessing.** We find the sequence length to be an important factor in achieving good performance. For all text classification datasets, we truncate the input to 512 subwords since BERT is pretrained with a maximum sequence length of 512. Further, when the length of an example is greater than 512, we keep the last 512 subwords instead of the first 512 subwords as keeping the latter part of the sentence lead to better performances on IMDb.

**Fine-tuning BERT on in-domain unsupervised data.** We fine-tune the BERT model on in-domain unsupervised data using the code released by BERT. We try learning rate of 2e-5, 5e-5 and 1e-4, batch size of 32, 64 and 128 and number of training steps of 30k, 100k and 300k. We pick the fine-tuned models by the BERT loss on a held-out set instead of the performance on a downstream task.

**Random initialized Transformer.** For the experiments with randomly initialized Transformer, we adopt hyperparameters for BERT base except that we only use 6 hidden layers and 8 attention heads. We also increase the dropout rate on the attention and the hidden states to 0.2, When we train UDA with randomly initialized architectures, we train UDA for 500k or 1M steps on Amazon-5 and Yelp-5 where we have abundant unlabeled data.

**BERT hyperparameters.** Following the common BERT fine-tuning procedure, we keep a dropout rate of 0.1, and try learning rate of 1e-5, 2e-5 and 5e-5 and batch size of 32 and 128. We also tune the number of steps ranging from 30 to 100k for various data sizes.

**UDA hyperparameters.** We set the weight on the unsupervised objective $\lambda$ to 1 in all of our experiments. We use a batch size of 32 for the supervised objective since 32 is the smallest batch size on v3-32 Cloud TPU Pod. We use a batch size of 224 for the unsupervised objective when the Transformer is initialized with BERT so that the model can be trained on more unlabeled data. We find that generating one augmented example for each unlabeled example is enough for BERT$_{\text{FINETUNE}}$.

All experiments in this part are performed on a v3-32 Cloud TPU Pod.

## E.2 Semi-supervised learning benchmarks CIFAR-10 and SVHN

**Hyperparameters for Wide-ResNet-28-2.** We train our model for 500K steps. We apply Exponential Moving Average to the parameters with a decay rate of 0.9999. We use a batch size of 64 for labeled data and a batch size of 448 for unlabeled data. The softmax temperature $\tau$ is set to 0.4. The confidence threshold $\beta$ is set to 0.8. We use a cosine learning rate decay schedule: $\cos(\frac{7t}{8T} * \frac{\pi}{2})$ where $t$ is the current step and $T$ is the total number of steps. We use a SGD optimizer with nesterov momentum with the momentum hyperparameter set to 0.9. In order to reduce training time, we generate augmented examples before training and dump them to disk. For CIFAR-10, we generate 100 augmented examples for each unlabeled example. Note that generating augmented examples in an online fashion is always better or as good as using dumped augmented examples since the model can see different augmented examples in different epochs, leading to more diverse samples. We report the average performance and the standard deviation for 10 runs. Experiments in this part are performed on a Tesla V100 GPU.

**Hyperparameters for Shake-Shake and PyramidNet.** For the experiments with Shake-Shake, we train UDA for 300k steps and use a batch size of 128 for the supervised objective and use a batch size of 512 for the unsuperivsed objective. For the experiments with PyramidNet+ShakeDrop, we train UDA for 700k steps and use a batch size of 64 for the supervised objective and a batch size of 128 for the unsupervised objective. For both models, we use a learning rate of 0.03 and use a cosine learning decay with one annealing cycle following AutoAugment. Experiments in this part are performed on a v3-32 Cloud TPU v3 Pod.

## E.3 ImageNet

**10% Labeled Set Setting.** Unless otherwise stated, we follow the standard hyperparameters used in an open-source implementation of ResNet.[6] For the 10% labeled set setting, we use a batch size of 512 for the supervised objective and a batch size of 15,360 for the unsupervised objective. We use a base learning rate of 0.3 that is decayed by 10 for four times and set the weight on the unsupervised objective $\lambda$ to 20. We mask out unlabeled examples whose highest probabilities across categories are less than 0.5 and set the Softmax temperature to 0.4. The model is trained for 40k steps. Experiments in this part are performed on a v3-64 Cloud TPU v3 Pod.

**Full Labeled Set Setting.** For experiments on the full ImageNet, we use a batch size of 8,192 for the supervised objective and a batch size of 16,384 for the unsupervised objective. The weight on the unsupervised objective $\lambda$ is set to 1. We use entropy minimization to sharpen the prediction. We use a base learning rate of 1.6 and decay it by 10 for four times. Experiments in this part are performed on a v3-128 Cloud TPU v3 Pod.

## Footnotes

[4]http://bit.ly/2kRWoof, https://ai.stanford.edu/~amaas/data/sentiment/

[5]https://www.kaggle.com/yelp-dataset/yelp-dataset, http://jmcauley.ucsd.edu/data/amazon/

[6]https://github.com/tensorflow/tpu/tree/master/models/official/resnet