[Reviews · NeurIPS 2020]

Review 1

Summary and Contributions: In this paper, the authors show that the quality of noising plays a crucial role in semi-supervised learning. The authors propose that using advanced data augmentation methods rather than simple noising improves performance on various text and vision related tasks. They use consistency loss to constrain the model predictions of unlabeled examples to be invariant to noise from data augmentations.

Strengths: The paper presents strong and extensive empirical results and outperform state of the art on different tasks. The paper is well written and easy to understand.

Weaknesses: The authors do not provide strong theoretical guarantees for why the method works. Its hard to establish the role the advanced data augmentation strategies play compared to the parameters/hyper-parameters used in tuning the models. The work is not novel as the various transformations applied to the data are well established. The authors apply them for semi-supervised learning so their effectiveness is not surprising.

Correctness: The empirical methodology is correct as the authors show the performance of their method of various tasks.

Clarity: The paper is clear and well written.

Relation to Prior Work: The authors have applied existing methods for a new problem and shown good results. In essence, the main difference to prior work is domain the methods are applied to.

Reproducibility: Yes

Additional Feedback: The main comment I have regarding the paper is that the authors do not provide adequate justification as to why the advanced data augmentation work compared to the simple ones and when to apply them. - The intuition provided in the theory is that unsupervised data augmentation will traverse the entire sub-graph of each disconnected component. This same intuition can be applied for other semi-supervised methods like nearest neighbor and label propagation. These methods will assign the same labels to unlabeled data examples within its component in a graph. - Assuming the noise eps is between (0,1), theorem 1 says that the error of UDA decreases as we get more labeled examples and the error increases as the noise increases. This is intuitive but does not explain why the noise from the advanced data augmentation methods are better for semi-supervised learning or provide guarantees for when they work. ========================================= I acknowledge that I read the rebuttal and thank the authors for providing explanations to the questions and concerns I had.


Review 2

Summary and Contributions: This paper introduces a generic method for semi-supervised learning to drastically improve the performance by applying strong data augmentation to unlabeled data and forcing the consistency of representation between such augmented data. The method is evaluated on both vision and language. The comprehensive experiments demonstrate the effectiveness of UDA.

Strengths: * The method does not depend on domain-specific data augmentation (e.g., mixup). Therefore, UDA can be applied to various domains, such as vision and language, as experimented in this paper. * The effectiveness of the method is comprehensively evaluated vision and language classification tasks, and UDA outperforms baselines in a large margin.

Weaknesses: * In terms of "valid noise" mentioned in L117, I think the adaptive variant of AutoAugment used in ReMixMatch [Berthelot et al. 2020] is more suitable for vision task. On the other hand, RandAugment used in UDA shares magnitude parameters, and some operations may be too strong or too weak, while others are appropriate. Berthelot et al. 2020 ReMixMatch: Semi-Supervised Learning with Distribution Matching and Augmentation Anchoring

Correctness: * Empirically, the method is well evaluated. * The theoretical analysis is overall sound, but I cannot understand what makes the following state correct: L 192, "performing unsupervised data augmentation ensures the traversal of the entire sub-graph C_i". The concern here is resolved in the rebuttal. The figures well explain the idea.

Clarity: The paper is well written and well organized.

Relation to Prior Work: The relation to prior work is well discussed. Yet, the reference is not well refined: for example, L472 Wide ResNet is accepted to BMVC, but the paper cites its arxiv version.

Reproducibility: Yes

Additional Feedback:


Review 3

Summary and Contributions: They have shown that in semi-supervised learning (SSK) tasks, the use of advanced data augmentation techniques such as RandAugment and back-translation can improve the performance of the consistency based training method. Through experiments, they show that the use of the augmentation policy can boost the performance compared to other methods and show that the technique is widely useful in vision and language tasks.

Strengths: 1. The simplicity of the method is favorable aspect. Then, as we have more advanced data augmentation technique, the scheme of the method should be applicable too. 2. They show that the proposed way is widely applicable, not limited to image classification task. It will attract the attention from many researchers of wide area.

Weaknesses: 1. While its simplicity, their contribution can be limited since they simply replaced the augmentation with recent state-of-the art data augmentation presented in other papers. 2. We could see that the use of RandAugment can improve the performance of SSL. But, which augmentation technique (cropping, random distortion...) was the key of improvement, or the use of all techniques was the key? If we also combine VAT with the proposed method, how the performance will be? This kind of analysis looks lacking and the empirical insight obtained from the paper is a little limited.

Correctness: The claims and method should be correct.

Clarity: Yes, the paper is well written and very easy to follow.

Relation to Prior Work: The relation is clearly discussed and their contribution is clear.

Reproducibility: Yes

Additional Feedback: I have a mixed thought on this paper. While great improvement on SSL task, the method does not look novel to me. If there is anything I missed in this paper, please point out it and respond to weaknesses I have shown. My concern was addressed after rebuttal and raised the score.


Review 4

Summary and Contributions: In this paper, the authors propose to use advanced data augmentation techniques in supervised learning as a superior source of noise in consistency training. The authors also provide a theoretical analysis of how UDA improves the classification performance and the corresponding role of the state-of-the-art augmentation. Empirical studies on a wide variety of language and vision tasks show significant improvements over state-of-the-art models with much less data.

Strengths: 1. Overall, the paper is well written and easy to follow. 2. It is an interesting idea to utilize data augmentation techniques in supervised learning as a superior source of data for consistency training. This idea is well motivated and both empirically and theoretically validated, making the method convincing. 3. Abundant and extensive experiments are conducted and the experimental results are really promising and encouraging.

Weaknesses: The most impressive point of this paper is its really perfect empirical results. However, it looks not surprising to me that superior data augmentations can benefit SSL since weak augmentation methods such as cropping have already been adopted in consistency training. Except for the insights from theoretical analysis, the novelty of the method seems to be limited since most of the techniques used are well established.

Correctness: The claims and methods are both theoretically and empirically validated.

Clarity: Yes. The paper is well formed and easy to follow.

Relation to Prior Work: Clear.

Reproducibility: Yes

Additional Feedback:

[Author Response · NeurIPS 2020]

We thank the reviewers for constructive feedback. We are delighted that the reviewers find the paper well-written and
appreciate the strong empirical results as well as the theoretical analysis.

*To Reviewer 1:* **[Intuition of benefits of advanced data augmentation]** In line 198, we explained the theoretical
connection between advanced data augmentation and better semi-supervised learning performance. We stated that
"Importantly, the number of components is actually decided by the quality of the augmentation operation: an ideal
augmentation should be able to reach all other examples of the same category given a starting instance. This well
matches our discussion of the benefits of state-of-the-art data augmentation methods in generating more diverse
examples. Effectively, the augmentation diversity leads to more neighbors for each node, and hence reduces the number
of components in a graph."

(a) Supervised learning (4/15)  (b) Advanced supervised augmentation (9/15)  (c) UDA with advanced augmentation (15/15)  (d) Simple supervised augmentation (7/15)  (e) UDA with simple augmentation. (10/15)

Figure 1: Prediction results of different algorithms, where green and red nodes are labeled nodes, white nodes are unlabeled nodes whose labels cannot be determined and light green nodes and light red nodes are unlabeled nodes whose labels can be correctly determined. The accuracies of different algorithms are shown in $(\cdot)$.

Since supervised data augmentation only propagates the label information to the directly connected neighbors of the
labeled nodes. Advanced data augmentation that has a high accuracy must lead to a graph where each node has more
neighbors. Effectively, such a graph has more edges and better connectivity. Hence, it is also more likely that this graph
will have a smaller numbers of components. To further illustrate this intuition, in Figure 1, we provide a comparison
between different algorithms. In contrast, the neighbors in nearest neighbor and label propagation are determined by
Euclidean distances, which may not have the same labels and may violate the label-preserving assumption used in our
analysis. We will include this detailed explanation in the future version.

*To Reviewer 2:* **[The traversal of the entire sub-graph]** The traversal means that consistency training can propagate
labels from labeled nodes to directly connected unlabeled nodes, and then to all connected unlabeled nodes in a
component. Please see Figure 1 for an illustration.

**[Citation style]** Thank you for pointing this out! We will refine the citation style in the future version.

**[Adaptive variant of AutoAugment]** We agree that adaptively refining the data augmentation can provide more valid
noise. We will include this insight in the future version.

*To Reviewer 3:* **[Contributions]** Our contributions are not only a simple change that leads to better performance, but
also an effective framework that is applicable to many tasks, the theoretical insight on why advanced data augmentation
works, the state-of-the-art performance and the consistency between theory and practice. As the reviewer has noted, the
proposed method is simple and widely applicable, which will attract attention from many researchers of different areas.

**[Lacking analysis]** We acknowledge that no ablation studies are included in the main paper due to the space limits.
The ablation studies are available in the supplementary material B.2. We show that the success of RandAugment should
be credited to the diversity of the augmentation transformations, since the model's performance gradually improves as
we use more augmentation transformations.

For effective augmentation techniques, if we only use one data augmentation technique, the best augmentations are
Equalize, Color and Brightness for CIFAR-10 and Invert, Equalize and ShearX for SVHN. We have also performed
experiments that combine UDA with VAT. We find that UDA+VAT leads to similar performance with UDA as shown in
Table 1, which means that the data augmentation noise is good enough and adding extra adversarial Gaussian noise
does not help.

| Methods / # Sup | 250 | 500 | 1,000 | 2,000 | 4,000 |
|---|---|---|---|---|---|
| UDA | $5.43 \pm 0.96$ | $4.80 \pm 0.09$ | $4.75 \pm 0.10$ | $4.73 \pm 0.14$ | $4.32 \pm 0.08$ |
| UDA + VAT | $5.89 \pm 1.12$ | $4.86 \pm 0.16$ | $4.81 \pm 0.13$ | $4.65 \pm 0.07$ | $4.27 \pm 0.15$ |

Table 1: Comparison between UDA and UDA + VAT

*To Reviewer 4:* We thank the reviewer for the feedback. We will clarify the our contributions in the future version.

[Meta-Review · NeurIPS 2020]

Four knowledgeable reviewers support acceptance for the contributions.Reviewers find the paper well-written and appreciate the strong empirical results as well as the theoretical analysis. Thr reviewers indicated that the method does not depend on domain-specific data augmentation (e.g., mixup). Therefore, UDA can be applied to various domains, such as vision and language, as experimented in this paper and it will attract the attention from many researchers of wide area. Therefore, I also recommend acceptance. However, please consider revising your paper to address all the concerns and comments from the reviewers.